# Production of Active Recombinant Hyaluronidase Inclusion Bodies from *Apis mellifera* in *E. coli* Bl21(DE3) and characterization by FT-IR Spectroscopy

**DOI:** 10.3390/ijms21113881

**Published:** 2020-05-29

**Authors:** Andreas Schwaighofer, Sarah Ablasser, Laurin Lux, Julian Kopp, Christoph Herwig, Oliver Spadiut, Bernhard Lendl, Christoph Slouka

**Affiliations:** 1FG Environmental Analytics, Process Analytics and Sensors, Institute of Chemical Technology and Analytics, Vienna University of Technology, Getreidemarkt 9/164, 1060 Wien, Austria; andreas.schwaighofer@tuwien.ac.at (A.S.); laurin.lux@tuwien.ac.at (L.L.); bernhard.lendl@tuwien.ac.at (B.L.); 2FG Bioprocess Technology, ICEBE, Vienna University of Technology, Gumpendorferstrasse 1a, 1060 Vienna, Austria; sarah.ablasser@hotmail.com (S.A.); julian.kopp@tuwien.ac.at (J.K.); christoph.herwig@tuwien.ac.at (C.H.); 3FG Integrated Bioprocess Development, ICEBE, Vienna University of Technology, Gumpendorferstrasse 1a, 1060 Vienna, Austria; oliver.spadiut@tuwien.ac.at

**Keywords:** *Escherichia coli*, active inclusion body, recombinant protein production, upstream process development, FT-IR spectroscopy

## Abstract

The bacterium *E. coli* is one of the most important hosts for recombinant protein production. The benefits are high growth rates, inexpensive media, and high protein titers. However, complex proteins with high molecular weight and many disulfide bonds are expressed as inclusion bodies (IBs). In the last decade, the overall perception of these IBs being not functional proteins changed, as enzyme activity was found within IBs. Several applications for direct use of IBs are already reported in literature. While fluorescent proteins or protein tags are used for determination of IB activity to date, direct measurements of IB protein activity are scacre. The expression of recombinant hyaluronidase from *Apis mellifera* in *E. coli* BL21(DE3) was analyzed using a face centered design of experiment approach. Hyaluronidase is a hard to express protein and imposes a high metabolic burden to the host. Conditions giving a high specific IB titer were found at 25 °C at low specific substrate uptake rates and induction times of 2 to 4 h. The protein activity of hyaluronidase IBs was verified using (Fourier transform) FT-IR spectroscopy. Degradation of the substrate hyaluronan occurred at increased rates with higher IB concentrations. Active recombinant hyaluronidase IBs can be immediately used for direct degradation of hyaluronan without further down streaming steps. FT-IR spectroscopy was introduced as a method for tracking IB activity and showed differences in degradation behavior of hyaluronan dependent on the applied active IB concentration.

## 1. Introduction

The bacterium *Escherichia coli* (*E. coli*) is one of the most extensively used prokaryotic organisms for recombinant protein production [1]. Being an expression host of choice for a long time, the genetic basis of *E. coli* is well characterized, and many manipulation tools are available [2]. Although *E. coli* lacks the post-translational machinery, advantages, like fast growth to high cell densities on comparatively inexpensive media, high biomass yields and simple scale-up, make the bacterium favorable [3]. Today, various eukaryotic expression systems are used to produce recombinant proteins, but *E. coli* accounts for the production of nearly 40% of all approved biopharmaceuticals [4]. Monoclonal antibodies and antibody fragments are currently the most important biopharmaceutical products. This further expands the use of *E. coli* as production host, as fragmented antigen binding antibodies (Fabs) can be successfully expressed [5,6]. 

The *E. coli* strain BL21(DE3), created by Studier and Moffatt back in 1986 [7], is often used for industrial production as it allows for high replication rates and shows low acetate formation [8]. In combination with the pET expression system, high transcriptional rates can be achieved [9]. Using such strong expression systems combined with harsh inducers, like IPTG, leads to high product yields, but also causes a high metabolic burden to the cells, often resulting in intracellular protein aggregates, called inclusion bodies (IBs). Inclusion bodies are insoluble protein aggregates, formerly considered as waste products [10]. Proteins accumulate due to specific stress reactions, like strong overexpression, high inducer concentrations, pH shifts, high temperatures and uptake rates, resulting in biologically inactive proteins [11]. Such IBs can be exploited for the production of toxic or unstable proteins that can subsequently be refolded in vitro [12]. Efficient refolding procedures are established today, but this time-consuming step to gain active protein for therapeutic use is a major drawback.

However in recent years, it was discovered that IBs contain a reasonable amount of correctly folded and thus biologically active protein and the misconception of IBs being inactive products composed of unfolded or misfolded proteins has changed [13,14,15]. Showing high specific activities suggests enrichment of active protein within IBs, making them important tools in the industrial biotechnological market and for biomedical applications [10,16,17]. Catalytically active IBs (CatIBs) as carrier-free protein immobilisates are promising biomaterials for synthetic chemistry, biocatalysis and biomedicine [18,19]. Furthermore, the correctly folded polypeptides within IBs coexist with an amyloid-like intermolecular beta-sheet structure conferring mechanical stability to IBs [20,21]. Thus, IBs can be useful systems for therapeutic approaches studying pathologic protein deposition in amyloid diseases, like Alzheimer’s or Parkinson’s disease, in which the accumulation of proteins initiates the pathogenic process [22,23,24,25]. 

Inclusion bodies also unveil high potential in biomedical applications in vivo as delivery vehicles or ‘nanopills’ for prolonged drug release [26,27,28,29]. Active IBs, as nanostructured amyloidal particles of 50-500 nm, are considered as mimetics of the endocrine secretory granules because they naturally penetrate mammalian cells and release their protein in soluble and functional form under physiological conditions [30]. It was shown, that decorating 3D-scaffolds with bacterial IBs favored mammalian cell surface colonization and stimulating proliferation and allowed the penetration and intracellular delivery of functional protein in absence of cytotoxicity and hence, offers possibilities in tissue engineering and regenerative medicine [31,32,33]. However, IBs are constrained for the use in biomedicine by their bacterial origin and undefined composition. To overcome these constraints, the creation of artificial IBs resulting in homogenous protein reservoirs for prolonged in vivo delivery of tumor-targeted drugs has been reported recently [34]. As new approaches emerge and the production of active IBs gains more interest, the impact of upstream process parameters on IB attributes becomes more important as they highly affect the amount of active protein within IBs [35].

Fourier transform-infrared (FT-IR) spectroscopy is a powerful analytical technique that provides molecule specific qualitative and quantitative information in a nondestructive and label-free manner by probing molecular vibrations [36]. It is a commonly used method for the analysis of biological samples, in particular proteins [37] and carbohydrates [38]. IR spectroscopy was successfully employed for monitoring enzyme activity by evaluating spectral changes of the substrate triggered by the enzymatic reaction [39,40,41]. Most recently, it was applied for secondary structure characterization of IBs [42,43,44,45]. To the best of our knowledge, this is the first report of assessing CatIB activity by IR spectroscopy. 

Hyaluronidase is an important enzyme for oncological care. The enzyme itself is known for rendering tissues more permeable for drug absorption (spreading effect) [46]. It is further used for defined production of hyaluronan oligosaccharides, which play crucial roles in different cellular responses. One of the main applications is tumor suppression and angiogenesis–formation of new blood vessels in tissues [47,48]. The goal of this study was the production of a high amount of active IBs of a hyaluronidase from *Apis mellifera* for direct application in the production of defined hyaluronan. Our hypotheses were (1) that adjustment of process parameters during the upstream process does not only affect productivity but also IB activity, and (2) that IR spectroscopy can be used to measure IB activity. For recombinant protein expression, *E. coli* BL21(DE3) was used as expression host in combination with the pET expression system and IPTG as inducing agent. In a fed batch cultivation, IB production was triggered in the induction phase. The upstream process parameters temperature and specific substrate uptake rate were altered using design of experiments (DoE) approaches and the IB size, and titer were analyzed. For activity assessment FT-IR analysis was used, and we showed that the produced hyaluronidase IBs exhibited reasonable activity and were able to degrade hyaluronan in situ. 

## 2. Results

### 2.1. Variation of Specific Substrate Uptake Rate and Temperature Using a DoE Approach

In order to check for multivariate dependencies, we cultivated *E. coli* harboring the target protein in a classical fed batch approach and altered temperature and specific substrate uptake rate (q_s,C_) in the induction phase based on the design in the Materials Part. Expression of recombinant hyaluronidase in *E. coli* led to a very high metabolic burden. Time-dependent cultivation responses are given in Figure 1. The four center point cultivations were compared in terms of volumetric recombinant protein titer and glycerol accumulation in the broth. Titer had a rather high standard deviation as starting biomass before induction differed in a range of approx. 25–30 g/L dry cell weight (DCW), however, the overall trend of recombinant protein expression could be clearly dedicated from the center point runs. Expression maximum was at very early induction times of approx. 2 to 4 h, with titers of 100 mg/L recombinant IBs. Longer expression led to high metabolic burden, which resulted in massive accumulation of the carbon source in the broth. This also resulted in a pronounced drop in specific growth rate and carbon uptake rate, often seen during harsh induction using IPTG [49]. Hence, biomass was only slowly growing during the induction phase and stopped after 2–4 h completely (Figure 1). In Figure 2 the statistical evaluation of different process and product parameters is presented. Figure 2a shows the maximal glycerol concentration in the broth in the induction phase. Irrespective of temperature, high accumulation occurred at medium to high specific substrate uptake set points, as growth rates decreased. A very similar dependence was found for acetate production (Figure 2b), which is only dependent on the applied q_s,C_.

Analogue responses were found for the protein titer after 4 h of induction (Figure 2c). This response was chosen, because the highest specific IB titers were found after 4 h of induction. Low specific substrate uptake rates in combination with low temperatures seem to be the optimal conditions for a high specific IB titer. Linear terms for temperature and quadratic terms for q_s,C_ are significant in the model. The linear q_s,C_ term is only close to significance, but kept in the model to increase the quality of the fit. Quadratic interaction could only be roughly estimated through this DoE model and did not reflect absolute values (compare to Appendix A). 

The averaged specific IB productivity (q_p,mean_) over the entire induction time showed only dependence on temperature (Figure 2d). Only positive values were considered, and degradation of the product was not taken into account. Considering the overall uncertainty of these values based on titer values/divided by biomass per time, the best induction strategy for production of recombinant hyaluronidase IBs is at low temperatures at a low specific substrate uptake rate for a maximum of 4 h of induction. The statistical evaluation can be found in Appendix A. 

We also analyzed IB bead size using scanning electron microscopy (SEM), for cultivations and induction times with the highest IB titers. 50 individual IB beads were measured on different images. It was found that the bead size was 230 nm throughout the analyzed cultivation with about 15% standard deviation. It was shown in recent studies by our group that IB size is strongly related to the IB titer during cultivation [44,50]. Both results indicate no new recombinant protein production after 4 h as also no increase in IB size could be observed. Figure 3a shows a representative SEM picture of hyaluronidase IBs of a centerpoint cultivation. Grey parts are host cell impurities, while bright spots are actual IBs. For purity determination we performed SDS-Page of two cultivation runs to compare purity during expression. The corresponding gel is given in Figure 3b. First line is the ladder, line 2 to 5 a commercially available BSA standard and lines 6 to 15 hyaluronidase samples. Titers of the IBs are below 0.5 g/L and therefore in good accordance with HPLC results. The MW of hyaluronidase is about 43 kDa with the clearly visible double peak as already known in literature [51]. The purity differs between 30 to 45% for the analyzed samples. As the IBs were only washed with ultrapure water, several host cell proteins can still be found in the pellet (which are also visible in the SEM). An extensive washing procedure with buffers including detergents like Triton X 100 may increase purity further as water insoluble host cell impurities will be solved.

### 2.2. Hyaluronidase IB Activity Measured by FT-IR Spectroscopy

As IR measurements demand higher overall concentration of recombinant protein, activity could not be analyzed of all experimental points of the DoE based on the relatively low titer. In order to produce high amounts of IBs, we used the optimal condition of 4 h induction at 25 °C and a q_s,C_ of 0.1 g/g/h and cultivated and harvested about 4 L of broth. IBs were centrifuged and re-suspended for IR measurements. The activity of the produced hyaluronidase IBs was tested by monitoring the enzymatic reaction with its native substrate, hyaluronan, by FT-IR spectroscopy. Hyaluronan is a polysaccharide of high molecular weight composed of repeating disaccharide units of D-glucuronic acid–β(1,3)-*N*-acetyl-D-glucosamine linked by β(1,4) glycosidic bonds. The hyaluronidase produced by *Apis mellifera* hydrolyses the glycosidic bonds in the β(1,4) position [45]. It was found that hyaluronidase exhibits an atypical Michaelis-Menten behavior at high substrate concentrations potentially because of non-specific complexes between hyaluronan and hyaluronidase based on electrostatic interactions. [52] Furthermore, the high viscosity of highly concentrated hyaluronan close to the solubility limit of 5 mg/mL may contribute to the atypical behavior due to steric exclusions of hyaluronidase from hyaluronan solution. Regarding the sensitivity of FT-IR spectroscopy, there is also a lower limit for the substrate concentration. For this reason, a hyaluronan concentration of 1 mg/mL was chosen for all IB activity test measurements.

The reaction of the IB-substrate solutions at IB concentrations between 1.5 and 30 mg/mL was monitored by FT-IR spectroscopy. Appendix A) shows a representative spectrum of the progression of the enzymatic reaction in the spectral region where the changes of the substrate occur. The band at 1077 cm^−1^ was assigned to the ν_C-O-C_ ring mode and the band at 1025 cm^−1^ could be attributed to the ν_C-OH_ modes of alcohols [53,54]. The band height of these two bands was evaluated and then the ratio was calculated. This normalization step was necessary, because the viscosity of the solution changed during the measurement, which lead to a changing baseline and varying absorbance across the entire spectrum [45]. For further analysis, the band height ratios were plotted versus the reaction time (Figure 4a). The obtained temporal progression was then fitted using a one-phase exponential grow function with time offset (with a high coefficient of determination, R^2^ > 0.98) and the growth rate was calculated as the inverse time constant. The growth rate shows a linear correlation to the IB concentration as visualized in Figure 4b.

## 3. Discussion

Hyaluronan oligosaccharides stimulate angiogenesis and suppress the growth of tumors [47]. Production of defined hyaluronan oligosaccharides is mandatory to fulfill this task. Active IBs of hyaluronidase are partly misfolded proteins, but exhibit residual protein activity, and are stable and easy to immobilize [55]. Such active IBs would facilitate degradation of hyaluronan to defined structures. Therefore, we cultivated *E. coli* expressing recombinant hyaluronidase, which shows rather low expression rates in *E. coli*. In order to increase the production of active IBs, we alternated the specific substrate uptake rate and induction temperature in a DoE design based on previous results [44]. We analyzed the expression in a small-scale system and expressed the protein in a larger scale to produce sufficient amounts of IB to comply with the sensitivity requirements of FT-IR spectroscopy, which we used to measure the activity of the IBs. Before cultivation we analyzed hyaluronidase from *Apis mellifera* with Expasy ProtPrameter server [56] and Disulfind server [57]. ProtParameter characterizes hyaluronidase as unstable with a computed instability index of 44.3 and Disulfind predicts two disulfide bonds with high probability. As the protein does not have a signal sequence for translocation into the periplasm, expression as IB is highly favored for this protein as disulfide bonds often cannot be correctly expressed in the cytoplasm. Recent works expressed hyaluronidase in *E. coli* as IBs in a shake flask-based approach with complex media. However, no upstream production in a controlled bioreactor system was performed [47,51,58]. The results of our cultivations showed expected expression as IBs, however, only in low concentrations. Cell stress is often the bottleneck for high protein expression in bacterial hosts. Recent studies marked several reasons for the so called metabolic burden upon recombinant protein expression [59,60,61,62,63,64]. Markers for metabolic burden during cultivation are a decrease of the specific growth rate (µ), resulting in glycerol accumulation and acetate formation.

However, such high acetate production is unexpected in BL21, as this strain is regarded as a low acetate former. This is based on an highly active glyoxylate shunt, which is generally inactive in the high acetate producing strain [65,66]. High acetate concentrations inhibit cell growth and recombinant protein production, which may be also the case in our cultivation. These high acetate concentrations indicate enormous metabolic stress levels upon production of recombinant hyaluronidase. Therefore, high uptake rates must be omitted. Low temperatures and specific glycerol uptake rates of 0.1 g/g/h fulfilled this task and were, therefore, well-chosen process parameters to successfully express hyaluronidase. 

Measurement of IB activity can be complex procedure depending on the protein of interest. Many groups use marker proteins, like GFP related fluorescence proteins, or protein activity assays to get insight into protein activity [10,11,67]. Within this work, initial tests of photometric assays using hyaluronan based upon sedimentation of particles or initial turbidity of the sample mix did only show significant results for commercial hyaluronidase, but not for the produced IBs (Sigma Aldrich test No: 3.2.1.35). Therefore, we aimed for an alternative measurement method for activity testing based on IR spectroscopy. As depicted in Figure 4b), the growth rate shows a linear correlation to the IB concentration, demonstrating and verifying the activity of the produced IB [68]. However, because of the applied normalization steps, no quantitative information about the enzyme velocity can be obtained. Comparison to commercially available hyaluronidase showed similar degradation behavior in FT-IR, given in Figure 5a. However different bands were evaluated for commercial enzyme compared to IBs. Details on the spectral information are given in Appendix A. For the commercial enzyme bands at 1113 cm^−1^ as ν_C-O-C_ glycosidic vibration and 1046 cm^−1^ as ν_C-OH_ vibration were used. Similar dependence on degradation behavior based on protein concentration is found for IBs and commercially available enzyme. However, absolute quantification based on the degradation curve is challenging, compare to Figure 5b.

Due to normalization of band changes absolute values are tricky to obtain and enzymatic activity depends on several parameters, such as pH and ionic strength [69,70]. However, based on a first photometric measurement, commercially derived enzymes seem to inherit a higher activity than the IB beads. For detailed enzymatic activity, the reaction products would have to be measured in a time dependent matter.

Nevertheless, it is shown that spectral changes caused by the enzyme reaction can be directly related to the enzyme activity, thus eliminating the need for further reaction steps that are usually required in conventional activity assays to obtain a reaction product detectable by UV/VIS spectroscopy. Planned improvements of this experimental approach, such as a temperature-controlled ATR manifold, will pave the way for a fully IR-based activity characterization of IBs and enzymes in situ. Alternatively, as IR spectroscopy was proven to be an excellent tool for in-line monitoring of bioprocesses by using fiber optic probes [71,72], IB activity measurements could also be performed in-line directly in an reactor vessel.

## 4. Materials and Methods 

### 4.1. Strains

*E. coli* BL21(DE3) was used with the pET28a plasmid system (kanamycin resistance) for recombinant protein production. A 1155 kb gene from *Apis mellifera* was codon optimized for *E. coli* and cloned at *Nde*I and X*ho*I into the pET plasmid by General Biosystems (Durham, NC, USA) and subsequently electroporated. Details on state-of-the-art cloning procedure are given in the Appendix A.

### 4.2. Bioreactor Cultivations

All preculture and bioreactor cultivations were carried out using a defined minimal medium referred to DeLisa et al. (1999) [73]. Batch media and the preculture media had the same composition with different amounts of glycerol—8 g/L for the preculture, 20 g/L for the batch phase. The feed for uninduced and induced fed-batch had a concentration of 400 g/L glycerol. Antibiotic was added throughout all fermentations in batch, resulting in a final concentration of 0.02 g/L of kanamycin. All precultures were performed using 500 mL high yield flasks. They were inoculated with 1.5 mL of bacteria solution stored in cryo stocks at −80 °C and subsequently cultivated for 20 h at 230 rpm in an Infors HR Multitron shaker (Infors, Bottmingen Switzerland) at 37 °C. IPTG was added once to start induction and had a final concentration of 0.5 mM. Recombinant protein production was performed in a DASGIP Mini bioreactor-4-parallel fermenter system (max. working volume: 2.5 L; Eppendorf, Hamburg, Germany). Cultivation offgas was analyzed by gas sensors-IR for CO_2_ and ZrO_2_ based for O_2_ (Blue Sens Gas analytics, Herten, Germany). For analytics, protein was produced in a stainless-steel Sartorius Biostat Cplus bioreactor (Sartorius, Göttingen, Germany) with 10 L working volume.

Process control was established using the PIMS Lucullus and the DAS-GIP-control system, DASware-control, which logged the process parameters. During batch-phase and fed-batch phase pH was kept constant at 6.7 and controlled with base only (12.5% NH_4_OH), while acid (5% H_3_PO_4_) was added manually, when necessary. The pH was monitored using an EasyFerm Plus pH-sensor (Hamilton, Reno, NV, USA). The reactors were continuously stirred at 1400 rpm and aerated using a mixture of pressurized air and pure oxygen at 2 vvm. Dissolved oxygen (dO_2_) was always kept higher than 30% by increasing the ratio of oxygen in the ingas. The dissolved oxygen was monitored using a fluorescence dissolved oxygen electrode Visiferm DO (Hamilton, Reno, NV, USA) The fed-batch phase for biomass generation was followed by an induction phase. Specific substrate uptake rate (q_s,C_) and temperature in the induction phase was adapted according to the DoE given in Figure 6. The specific substrate uptake rate was altered between 0.1 g/g/h and 0.5 g/g/h and temperature between 25 °C and 35 °C. The center point at 30 °C and 0.3 g/g/h was cultivated four times in order to assess the statistical experimental error. 

### 4.3. Cultivation Analytics

#### 4.3.1. Biomass

For dry cell weight (DCW) measurements 1 mL of the cultivation broth was centrifuged at 9000× *g*, subsequently washed with 0.9% NaCl solution and centrifuged again under the same conditions. After drying the cells at 105 °C for 48 h, the pellet was evaluated gravimetrically. DCW measurements were performed in five replicates and the mean error for DCW was approx. 3%. Offline OD_600_ measurements were performed in duplicates in a UV/VIS photometer Genisys 20 (Thermo Scientific, Waltham, MA, USA). 

#### 4.3.2. Sugar Analytics

Sugar concentrations in the filtered fermentation broth were determined using a Supelco C-610H HPLC column (Supelco, Bellefonte, PA, USA) on an Ultimate 3000 HPLC system (Thermo Scientific, Waltham, MA, US) using 0.1% H_3_PO_4_ as running buffer at 0.5 mL/min or an Aminex HPLC column (Biorad, Hercules; CA, USA) on an Agilent 1100 System (Agilent Systems, Santa Clara, CA, USA) with 4 mM H_2_SO_4_ as running buffer at 0.6 mL/min.

### 4.4. Product Analytics

#### 4.4.1. IB Preparation

5 mL fermentation broth samples were centrifuged at 4800 rpm at 4 °C for 10 min. The supernatant was discarded and the pellet was resuspended to a DCW of about 4 g/L in lysis buffer (100 mM Tris, 10 mM EDTA at pH = 7.4). Afterwards the sample was homogenized using a high-pressure homogenizer at 1500 bar for 10 passages (PandaPLUS, Gea AG, Germany). After centrifugation at 10,000 rpm and 4 °C the supernatant was discarded and the resulting IB pellet was washed twice with ultrapure water and aliquoted into pellets of 2 mL broth, centrifuged (14,000 rpm, 10 min 4 °C) and stored at −20 °C.

#### 4.4.2. IB Size

Washed and aliquoted IB samples were resuspended in ultrapure water. 100 µL of appropriate dilution of the suspension were pipetted on a gold-sputtered (10–50 nm) polycarbonate filter (Millipore-Merck, Darmstadt, Germany) using reusable syringe filter holders with a diameter of 13 mm (Sartorius, Göttingen, Germany). 100 µL of ultrapure water were added and pressurized air was used for subsequent filtration. Additional 200 µL of ultrapure water were used for washing. The wet filters were fixed on a SEM sample holder using graphite adhesive tape and subsequently sputtered with gold to increase the contrast of the sample. SEM was performed using a QUANTA FEI SEM (Thermo Fisher, Waltham, MA, USA) with a secondary electron detector [60]. The acceleration voltage of the electron beam was set between 3 to 5 kV. To determine the diameter of the IBs, 50 IBs on SEM pictures were measured using the ImageJ plugin Fiji (Laboratory for Optical and Computational Instrumentation (LOCI), University of Wisconsin-Madison, USA). SEM analytics of two different time points for both strains are given in Figure 2.

#### 4.4.3. IB Titer 

For titer measurements, IB pellets were solubilized using solubilization buffer (7.5 M guanidine hydrochloride, 62 mM Tris at pH = 8). The filtered samples were quantified by HPLC analysis. The HPLC was equipped with a BioResolve RP mAb Polyphenyl column (dimensions 100 × 3 mm, particle size 2.7 µm) which is designed for mAb and large protein analyses (Waters Corporation, MA, USA). To prolong column lifetime, a pre-column (3.9 × 5mm, 2.7 µm) was used. The mobile phase was composed of ultrapure water (=MQ, eluent A) and acetonitrile (eluent B) both supplemented with 0.10% (*v/v*) trifluoroacetic acid. The injection volume was set to 2.0 µL. The flow rate and the column temperature of the final method were 0.4 mL/min and 70 °C, respectively. Details on the measurement methodology is given in [74].

#### 4.4.4. SDS-PAGE

For the SDS-page Mini-PROTEAN^®^ TGX Stain-Free™ (BioRad, Hercules, CA, USA) was used. 4× Laemmli sample buffer (BioRad) containing 200 µL ß-mercaptoethanol was diluted 2 times with ultrapure water for the soluble samples and 4 times in order to dissolve the IB pellets. BSA standard with concentrations of 2, 1, 0.5, 0.1 g/L were mixed 1:1 with Laemmli buffer. The IB pellets were dissolved with 500 µL Laemmli buffer. All samples were denatured for 5 min at 95 °C and subsequently spun down. 5 µL protein standard (precision plus protein standard dual color, BioRad) and 10 µL of each sample were loaded onto the gel and it was run for 30 min at 180V. Afterwards the gel was stained for about an h using Coomassie brilliant blue (BioRad). 

#### 4.4.5. FT-IR Spectroscopy 

FT-IR absorption measurements of IB activity monitoring was performed using a Bruker Vertex 70V FT-IR spectrometer (Ettlingen, Germany) equipped with a Bruker Optics Platinum ATR module (diamond crystal, 1 mm^2^ area with single reflection) and a liquid nitrogen cooled HgCdTe (mercury cadmium telluride) detector. To start the enzymatic reaction, 15 µL of IB at different concentrations were mixed with 15 µL of 1 mg/mL hyaluronan (sodium salt from *Streptococcus equi*, bacterial glycosaminoglycan polysaccharide, No. 53747, Sigma Aldrich) and thoroughly vortexed. The sample solution was then placed on the ATR crystal and covered with a cap to prevent solvent evaporation during spectra acquisition. ATR-FT-IR spectra were collected every minute for a time period of 40 min. For IB concentration determination, commercially obtained hyaluronidase type I-S from bovine testes (No. H3506, Sigma Aldrich) was used for calibration and the amide II band height was evaluated. Spectra were acquired with a spectral resolution of 4 cm^−1^ in double-sided acquisition mode; the mirror velocity was set to 80 kHz. A total of 450 scans were averaged per spectrum, which was calculated using a Blackman-Harris 3-term apodization function and a zero-filling factor of 2. All spectra were acquired at 25 °C. Spectra were analyzed using the software package OPUS 8.1 (Bruker, Ettlingen, Germany).

## 5. Conclusions

Here we present a cultivation strategy to obtain a high specific titer of active hyaluronidase from *Apis mellifera* IBs in *E. coli* as well as a novel tool to determine their catalytic activity by using ATR FT-IR spectroscopy. We demonstrated that active hyaluronidase IB can be used for degradation of hyaluronan. Further evaluation of critical quality attributes will be necessary, but these active IBs may be a useful tool for pharmaceutical applications.

## Figures and Tables

**Figure 1 ijms-21-03881-f001:**
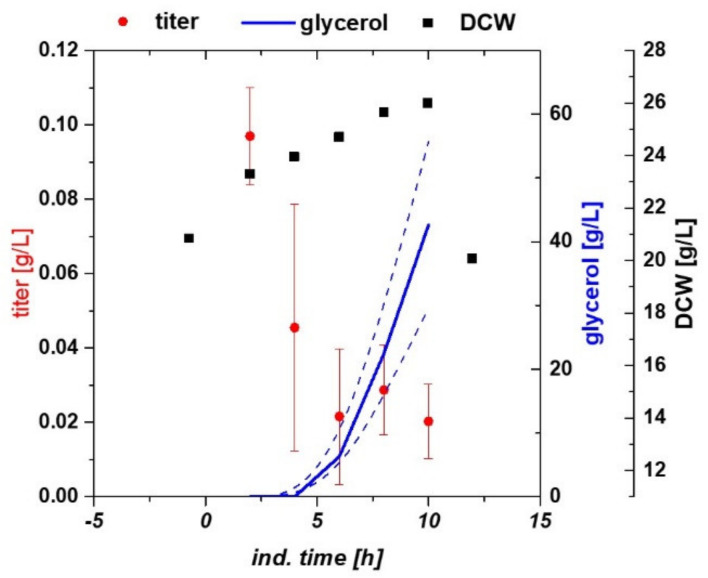
Centerpoint cultivations of *E. coli* expressing recombinant hyaluronidase. Recombinant protein expression is the highest at early induction times. As the protein is obviously hard to express, biomass growth decreases and glycerol accumulates in the broth over induction time.

**Figure 2 ijms-21-03881-f002:**
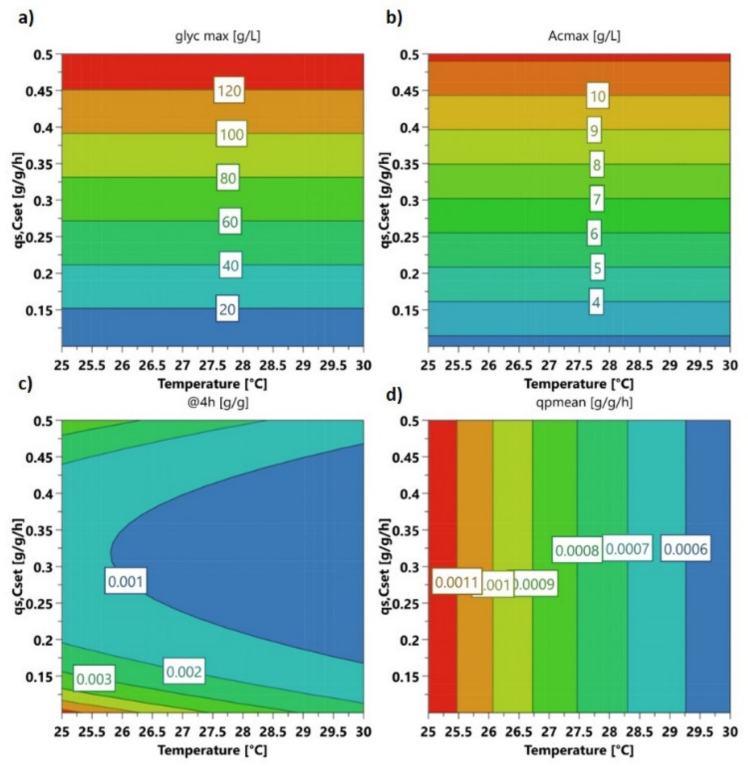
Contour plots. (**a**) maximum glycerol concentration in broth; (**b**) maximum acetate concentration in broth; (**c**) specific hyaluronidase IB titer at 4-h induction time; (**d**) mean specific productivity over the whole induction time.

**Figure 3 ijms-21-03881-f003:**
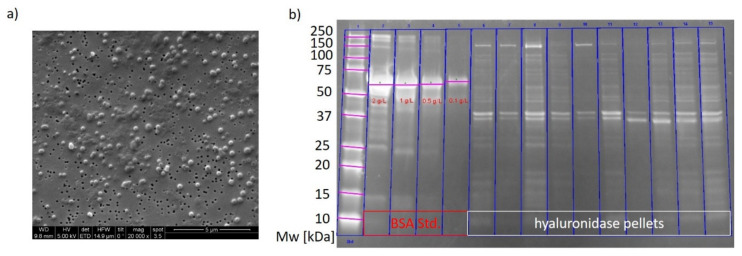
(**a**) SEM of hyaluronidase IBs. The bright spots are IB beads, while dark gray spots are host cell related impurities after the washing procedure; (**b**) SDS-PAGE of two hyaluronidase cultivations. Lane 1 is the molecular weight ladder, lane 2 to 5 the BSA standard, lane 6–10 and lane 11–15 two respective IB samples from two cultivations with sampling times of 4, 6, 8, 10 and 12 h of induction

**Figure 4 ijms-21-03881-f004:**
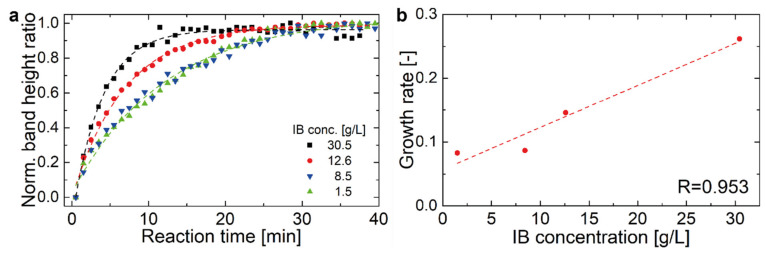
(**a**) Progression of the normalized band ratio with time of four investigated IB concentrations. (**b**) Growth rate vs IB concentration shows linear relationship.

**Figure 5 ijms-21-03881-f005:**
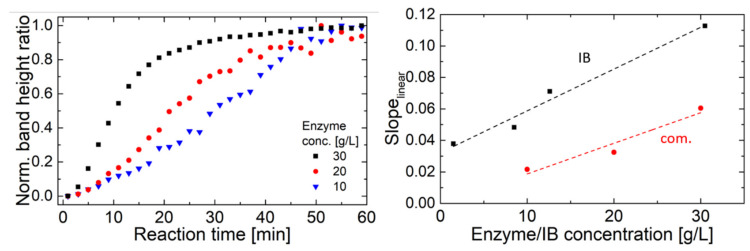
(**a**) degradation behavior of commercially available enzyme, (**b**) linear part of the slopes of the degradation curves vs. concentration of the enzymes.

**Figure 6 ijms-21-03881-f006:**
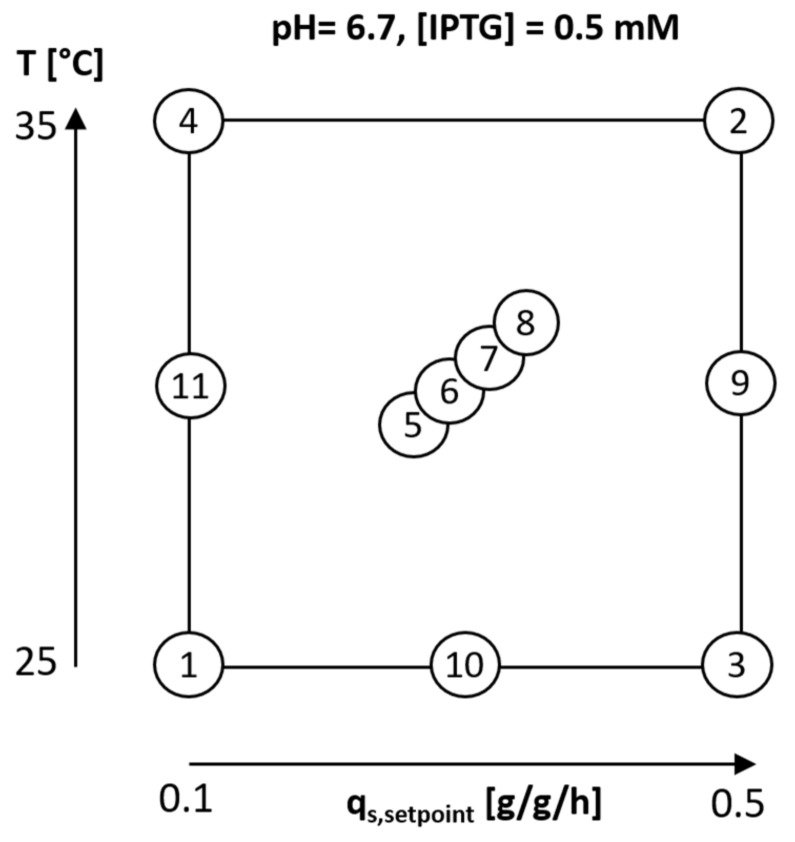
DoE for process optimization to produce active recombinant hyaluronidase IBs. Experiments 1–7 were performed in a modified CCF design. Additional experiments 9–11 were subsequently performed to analyze for quadratic interactions in the region of interest.

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
