# Peer review of "Production of Active Recombinant Hyaluronidase Inclusion Bodies from Apis mellifera in E. coli Bl21(DE3) and characterization by FT-IR Spectroscopy"

_ijms, 2020, doi:10.3390/ijms21113881_

Round 1

Reviewer 1 Report

In this manuscript by Schwaighofer et al, the authors have produced recombinant Hyaluronidase in E. coli BL21(DE3) from Inclusion Bodies and evaluated the enzymatic activity by FT-IR spectroscopy.

1) In my opinion the active Inclusion bodies should be evaluated for any contaminants derived for E.coli production for example SDS-PAGE analysis may be useful.

2) In the conclusion section the authors write: “We demonstrated that hyaluronidase IBs can be directly used for degradation of hyaluronan and may be a useful tool for pharmaceutical applications.” In my experience the only enzymatic activity as characterization product it is not enough to be used for pharmaceutical applications.

3) In my opinion is important (if possible) that the authors compare the activity of recombinant Hyaluronidase from IBs with a commercial enzyme.

4) In the section: 4.1. IB Preparation, the authors sometimes enter the minutes of centrifugation and sometimes not. please check it.

Author Response

In this manuscript by Schwaighofer et al, the authors have produced recombinant Hyaluronidase in E. coli BL21(DE3) from Inclusion Bodies and evaluated the enzymatic activity by FT-IR spectroscopy.

1) In my opinion the active Inclusion bodies should be evaluated for any contaminants derived for E. coli production for example SDS-PAGE analysis may be useful.

We added following part for purity determination:

50 individual IB beads were measured on different images. It was found that the bead size was in mean 230 nm throughout the analyzed cultivation with about 15 % standard deviation. It was shown in recent studies by our group that IB size is strongly related to the IB titer during cultivation [42,45]. Both results indicate no new recombinant protein production after 4 hours as also no increased in IB size could be observed. Figure 3a shows a representative SEM picture of hyaluronidase IBs of a centerpoint cultivation. Grey parts are host cell impurities, while bright spots are actual IBs. For purity determination we performed SDS-Page of two cultivation runs to compare purity during expression. The corresponding gel is given in Figure 3b. First line is the ladder, line 2 to 5 a commercially available BSA standard and lines 6 to 15 hyaluronidase samples. Titers of the IBs are below 0.5 g/L and therefore in good accordance with HPLC results. The MW of about hyaluronidase is about 43 kDa with the clearly visible double peak as already known in literature [46]. The purity differs between 30 to 45 % for the analyzed samples. As the IBs were only washed with ultrapure water, several host cell proteins can still be found in the pellet (which are also visible in the SEM). An extensive washing procedure with buffers including detergents like Triton-100 may increase purity further as water insoluble host cell impurities will be

2) In the conclusion section the authors write: “We demonstrated that hyaluronidase IBs can be directly used for degradation of hyaluronan and may be a useful tool for pharmaceutical applications.” In my experience the only enzymatic activity as characterization product it is not enough to be used for pharmaceutical applications.

We eased this statement and changed to following part:

We demonstrated that active hyaluronidase IB can be used for degradation of hyaluronan. Further evaluation of critical quality attributes will be necessary, but these active IBs may be a useful tool for pharmaceutical applications.

3) In my opinion is important (if possible) that the authors compare the activity of recombinant Hyaluronidase from IBs with a commercial enzyme.

The reviewer is completely right about this point, and we did several measurements on commercially available hyaluronidase. However, a direct comparison is rather challenging. During photometric assay determination, IBs did not show detectable activity, while commercial product reacted perfectly. For this reason, we tried FT-IR as more sensitive method. With IR we see the degradation for both enzymes (IBs and commercial), however direct comparison became impossible due to several reasons stated in the text. We believe that the commercial enzyme is more active, but IBs have several benefits as stated in the manuscript. We added following part:

Comparison to commercially available hyaluronidase showed similar degradation behavior in FT-IR. However different bands were evaluated for commercial enzyme compared to IBs. Details on the degradation behavior are given in Supplementary Figure 3. Due to normalization of band changes absolute values are tricky to obtain and enzymatic activity depend several parameters, like pH and ionic strength {Lenormand, 2013 }{Lenormand, 2011}. However, based on first photometric measurement, commercially derived enzymes seem to inherit a higher activity than the IB beads. For detailed enzymatic activity, the reaction products would have to be measured on a time dependent matter.

Nevertheless, it is shown that spectral changes caused by the enzyme reaction can be directly related to the enzyme activity, thus eliminating the need for further reaction steps that are usually required in conventional activity assays to obtain a reaction product detectable by UV/VIS spectroscopy. Minor comments:

4) In the section: 4.1. IB Preparation, the authors sometimes enter the minutes of centrifugation and sometimes not. please check it.

Added accordingly

Reviewer 2 Report

This is a very clearly written report on the production of inclusion bodies that contain Haluronidase a commercially interesting enzyme that degrades Haluronan. The optimal conditions for the production of the inclusion bodies was determined and the activity of the corrected folded Haluronidase in the inclusion bodies was assessed by IR spectroscopy.

The inclusion bodies seem to be fairly homogenously based on SEM images. And the activity of the enzyme correlates well with the concentration of inclusion bodies added to the reaction.  The authors suggest that the inclusion bodies can be used for pharmaceutical purposes. Possibly true, but the suggestion would be supported if the reader would know how pure the inclusion bodies are. Do they include many contaminations or are they relatively pure? Another interesting aspect would be the knowledge on amount of Haluronidase that functions as an active enzyme in the inclusion bodies. Is this 1% of 10%, 50%? A commercial enzyme is available and used by the authors, so probably they could make this comparison relatively easy.

Minor comments:

Please write DCW first time in full.

Line 145: Please mention the SD of the mean bead size and the number of beads measured.

Line 208: Please have a look at this sentence: However, such high acetate production is unexpected in BL21, as this strain is regarded as a low acetate former due to the active glyoxylate shunt in the low acetate producer, which is inactive in the high acetate producer [61,62].

Line 214: Task… were, make better connection.

Line 256: (12.5% NH4OH), while acid (5% H3PO4), not subscripts…also line 282, line 283

Author Response

This is a very clearly written report on the production of inclusion bodies that contain Haluronidase a commercially interesting enzyme that degrades Haluronan. The optimal conditions for the production of the inclusion bodies was determined and the activity of the corrected folded Haluronidase in the inclusion bodies was assessed by IR spectroscopy.

The inclusion bodies seem to be fairly homogenously based on SEM images. And the activity of the enzyme correlates well with the concentration of inclusion bodies added to the reaction.  The authors suggest that the inclusion bodies can be used for pharmaceutical purposes. Possibly true, but the suggestion would be supported if the reader would know how pure the inclusion bodies are. Do they include many contaminations or are they relatively pure? Another interesting aspect would be the knowledge on amount of Haluronidase that functions as an active enzyme in the inclusion bodies. Is this 1% of 10%, 50%? A commercial enzyme is available and used by the authors, so probably they could make this comparison relatively easy.

For purity, we performed SDS Page and found a purity of about 40%. This is not unexpected since overall titers are very low and the IB size is very small, resulting in a high surface to volume ratio. Furthermore, we only performed washing with MQ. A better washing procedure may boost purity to far higher values. We commented on this in following part and added Figure 3b.

50 individual IB beads were measured on different images. It was found that the bead size was in mean 230 nm throughout the analyzed cultivation with about 15 % standard deviation. It was shown in recent studies by our group that IB size is strongly related to the IB titer during cultivation [42,45]. Both results indicate no new recombinant protein production after 4 hours as also no increased in IB size could be observed. Figure 3a shows a representative SEM picture of hyaluronidase IBs of a centerpoint cultivation. Grey parts are host cell impurities, while bright spots are actual IBs. For purity determination we performed SDS-Page of two cultivation runs to compare purity during expression. The corresponding gel is given in Figure 3b. First line is the ladder, line 2 to 5 a commercially available BSA standard and lines 6 to 15 hyaluronidase samples. Titers of the IBs are below 0.5 g/L and therefore in good accordance with HPLC results. The MW of about hyaluronidase is about 43 kDa with the clearly visible double peak as already known in literature [46]. The purity differs between 30 to 45 % for the analyzed samples. As the IBs were only washed with ultrapure water, several host cell proteins can still be found in the pellet (which are also visible in the SEM). An extensive washing procedure with buffers including detergents like Triton-100 may increase purity further as water insoluble host cell impurities will be solved.

As IBs are just the catalyst and should be not soluble, we think that the overall purity is not of very high importance. The defined hyaluronan will be down streamed after reaction. However, endotoxins from E. coli may pose problems and leaches from IB may also be problematic.

We also did measurements with commercially available hyaluronidase. The degradation behavior is highly similar, however, different bands in the IR spectrum had to be evaluated as given in the new Figure in the Supplementary Part. The normalization of the spectrum makes it quantification furthermore challenging. We also found several publications, which show that enzymatic tests for hyaluronan are quite challenging. To check for conversation, the amount of reacted hyaluronan would have to be measured, which was out of the scope of this paper. Based on commercially available photometric tests, we did in first place before FT-IR, be believe that the commercial hyaluronidase is far more active. However, IBs have the benefit of being stable systems as the do are a suspension and may be easily immobilized, despite their lower activity.

Comparison to commercially available hyaluronidase showed similar degradation behavior in FT-IR. However different bands were evaluated for commercial enzyme compared to IBs. Details on the degradation behavior are given in Supplementary Figure 3. Due to normalization of band changes absolute values are tricky to obtain and enzymatic activity depend several parameters, like pH and ionic strength {Lenormand, 2013}{Lenormand, 2011}. However, based on first photometric measurement, commercially derived enzymes seem to inherit a higher activity than the IB beads. For detailed enzymatic activity, the reaction products would have to be measured on a time dependent matter.

Nevertheless, it is shown that spectral changes caused by the enzyme reaction can be directly related to the enzyme activity, thus eliminating the need for further reaction steps that are usually required in conventional activity assays to obtain a reaction product detectable by UV/VIS spectroscopy. Minor comments:

Please write DCW first time in full.

Changed accordingly

Line 145: Please mention the SD of the mean bead size and the number of beads measured.

Added following part:

50 individual IB beads were measured on different images. It was found that the bead size was in mean 230 nm throughout the analyzed cultivation with about 15 % standard deviation.

Line 208: Please have a look at this sentence: However, such high acetate production is unexpected in BL21, as this strain is regarded as a low acetate former due to the active glyoxylate shunt in the low acetate producer, which is inactive in the high acetate producer [61,62].

Changed the part to clarify the message:

However, such high acetate production is unexpected in BL21, as this strain is regarded as a low acetate former. This is based on an highly active glyoxylate shunt, which is generally inactive in the high acetate producing strain [61,62].

Line 214: Task… were, make better connection.

Changed the sentence accordingly:

Therefore, high uptake rates must be omitted. Low temperatures and specific glycerol uptake rates of 0.1 g/g/h fulfilled this task and were, therefore, well chosen process parameters to successfully express hyaluronidase.

Line 256: (12.5% NH4OH), while acid (5% H3PO4), not subscripts…also line 282, line 283

Changed all to subscripts

Round 2

Reviewer 1 Report

no comments

Author Response

We thank the reviewer very much!

Reviewer 2 Report

Minor comments:

Line 146: change to: 50 individual IB beads were measured on different images. It was found that the mean of the bead size was 230 nm throughout the analyzed cultivation with about 15 % standard deviation.

Line 148:change to:Both results indicate no new recombinant protein production after 4 hours as also no increase in IB size could be observed

Line 155: change to: The MW of hyaluronidase is about 43 kDa with the clearly visible double peak as already known in literature [46]

Line 240 change to: enzymatic activity depends on several parameters,

line 241 chnage to: based on a first photometric measurement

line 243: reaction products would have to be measured in a time dependent manner

Author Response

We thank the reviewer for the time spent to improve the quality of the manuscript! We marked the changes in the manuscript in red.

Line 146: change to: 50 individual IB beads were measured on different images. It was found that the mean of the bead size was 230 nm throughout the analyzed cultivation with about 15 % standard deviation.

Changed accordingly.

Line 148:change to:Both results indicate no new recombinant protein production after 4 hours as also no increase in IB size could be observed

Changed accordingly.

Line 155: change to: The MW of hyaluronidase is about 43 kDa with the clearly visible double peak as already known in literature [46]

Changed accordingly.

Line 240 change to: enzymatic activity depends on several parameters,

Changed accordingly.

line 241 chnage to: based on a first photometric measurement

Changed accordingly.

line 243: reaction products would have to be measured in a time dependent manner

Changed accordingly.